# Characterization of Microstructure, Optical Properties, and Mechanical Behavior of a Temporary 3D Printing Resin: Impact of Post-Curing Time

**DOI:** 10.3390/ma17071496

**Published:** 2024-03-26

**Authors:** Joyce Roma Correia dos Santos Siqueira, Rita Maria Morejon Rodriguez, Tiago Moreira Bastos Campos, Nathalia de Carvalho Ramos, Marco Antonio Bottino, João Paulo Mendes Tribst

**Affiliations:** 1Department of Dental Materials and Prosthodontics, São Paulo State University (Unesp), Institute of Science and Technology, São José dos Campos 12245-000, Brazil; joyce.roma@unesp.br (J.R.C.d.S.S.); rita.morejon@unesp.br (R.M.M.R.); nathalia.ramos@unesp.br (N.d.C.R.); marco.bottino@unesp.br (M.A.B.); 2Department of Prosthodontics and Periodontology, Bauru School of Dentistry, University of São Paulo, Bauru 17012-901, Brazil; moreiratiago22@gmail.com; 3Department of Reconstructive Oral Care, Academic Centre for Dentistry Amsterdam (ACTA), Universiteit van Amsterdam and Vrije Universiteit Amsterdam, 1081 LA Amsterdam, The Netherlands

**Keywords:** 3D print, 3D printed resins, additive manufacturing, dental materials, monomers

## Abstract

The present study aimed to characterize the microstructure of a temporary 3D printing polymer-based composite material (Resilab Temp), evaluating its optical properties and mechanical behavior according to different post-curing times. For the analysis of the surface microstructure and establishment of the best printing pattern, samples in bar format following ISO 4049 (25 × 10 × 3 mm) were designed in CAD software (Rhinoceros 6.0), printed on a W3D printer (Wilcos), and light-cured in Anycubic Photon for different lengths of time (no post-curing, 16 min, 32 min, and 60 min). For the structural characterization, analyses were carried out using Fourier transform infrared spectroscopy (FT-IR) and scanning electron microscopy (SEM). The mechanical behavior of this polymer-based composite material was determined based on flexural strength tests and Knoop microhardness. Color and translucency analysis were performed using a spectrophotometer (VITA Easy Shade Advanced 4.0), which was then evaluated in CIELab, using gray, black, and white backgrounds. All analyses were performed immediately after making the samples and repeated after thermal aging over two thousand cycles (5–55 °C). The results obtained were statistically analyzed with a significance level of 5%. FT-IR analysis showed about a 46% degree of conversion on the surface and 37% in the center of the resin sample. The flexural strength was higher for the groups polymerized for 32 min and 1 h, while the Knoop microhardness did not show a statistical difference between the groups. Color and translucency analysis also did not show statistical differences between groups. According to all of the analyses carried out in this study, for the evaluated material, a post-polymerization time of 1 h should be suggested to improve the mechanical performance of 3D-printed devices.

## 1. Introduction

Additive manufacturing (AM), also known as 3D printing or rapid prototyping, involves the fabrication of an object through the sequential application of thin layers of a material specified by a design executed in 3D modeling software in a process known as computer-aided design (CAD) [1,2], or from data obtained from computed tomography (CT), cone-beam computed tomography (CBCT), or scanning [3].

This type of printing technology originated in the 1980s and was originally used for the production of prototypes, models, and casting patterns [4], based on the patent applied for stereolithography printing acquired by Chuck W. Hull [5]. Following the first patent, various technologies were developed. In 2009, the American Society for Testing and Materials (ASTM), which regulates techniques and standards for a wide range of materials, products, systems, and services, defined seven categories for all existing 3D printing technologies, as described in the ISO/ASTM 52900-15 [6] standard: stereolithography (STL), material jetting (MJ), binder jetting, powder bed fusion (SLS), sheet lamination, and direct energy deposition [7]. These technologies can also be classified according to their material layering method, such as stereolithography, selective laser sintering, digital light processing, PolyJet, and fused deposition modeling, which are applied in the fabrication of dental prostheses [8,9,10]. 

The most widely used printers in dentistry are SLA and DLP types, derived from the technique used, in which the printing platform is immersed in resin polymerized by ultraviolet (UV) or LED (a set of lamps) light. The laser/LED draws a cross-section of the object to form each layer, repeating this process numerous times until the final construction of the printed piece is completed [3,5,11]. Various materials such as plastic, metal, ceramic, and polymers, among others, can be applied to this processing technique, which has several advantages over subtractive methods (milling), such as reduced material waste and the ability to construct complex geometric structures [8].

3D printing has numerous applications and has been implemented in various fields, such as the medical and dental fields. In medicine, the technology is used in orthopedics, neurosurgery, cardiac surgery, maxillofacial surgery, and other specialties [12]. In dentistry, this technology has been employed to eliminate numerous technical steps necessary for the fabrication of prosthetic restorations [13,14], as well as to enable the creation of study models, prototypes of anatomical structures to facilitate diagnosis and treatment planning, surgical guides, occlusal splints, orthodontic appliances, and permanent and temporary restorations [3].

The techniques used for the fabrication of temporary restorations can be divided into a direct and an indirect method, according to the manufacturing process. In the direct method, restorations are immediately fabricated on tooth preparations, while in the indirect method, crowns are manufactured from gypsum models or files obtained from intraoral scanning, and then installed on tooth preparations. Although the direct method is faster, it has more disadvantages, such as the fact that it is an exothermic reaction, and excessive heat released during the resin polymerization process can cause thermal trauma to the dental pulp. The residual monomer in the polymer used in the direct fabrication method can also injure the oral mucosa, causing blisters or allergic stomatitis. Additionally, there is undesirable polymerization shrinkage of the resin caused by the reduction of atomic distance in the low molecular weight of the monomers used, leading to dimensional changes in the marginal, occlusal, and interproximal regions. The indirect method, on the other hand, eliminates the risks of thermal and chemical reactions to the tooth and mucosa. Crown adaptation to the tooth is increased because the polymerization process is carried out extra-orally. Thus, it is essential to observe the characteristics of each technique in the selection of a material, such as working time, ease of fabrication and repair, biocompatibility, dimensional stability during and after fabrication, availability, and color stability [15]. Various materials for temporary restorations are available, such as acrylic resin, which is commonly used for its cost-effectiveness, aesthetic acceptance, and versatility, although it presents issues in the form of tissue toxicity and thermal irritation. Another material used is bis-acrylic resin, which was introduced to reduce issues with conventional acrylic resin. More recently, the use of 3D printing resins has been implemented, capable of faithfully reproducing restorations with precise dimensions and reduced fabrication time [16,17].

The recent use of 3D printers for obtaining temporary restorations has shown that the mechanical behavior of these resins is satisfactory [18]. Other studies have analyzed repair options, showing that surface treatments increase the bond strength between these 3D printing resins and conventional resins (such as bis-acrylic, methyl methacrylate-based, PMMA, or Bis-GMA), demonstrating that repair is possible in case of intraoral failure. However, studies on adhesive strength to dental substrate or cement are still limited, requiring further investigation into the 3D printing resins. This study aims to understand a 3D printing polymer-based composite material through microstructural, mechanical, and optical characterization. The null hypothesis is that there is no significant difference in the microstructure, optical properties, and mechanical behavior of a polymer-based composite material when subjected to different post-curing times.

## 2. Materials and Methods

### 2.1. Sample Manufacturing

The samples were designed in 3D modeling software (Rhinoceros 6.0) according to the selected design for each test (disk, square, and bars). The generated file was exported in STL format to the slicing software of the Anycubic printer (Anycubic, Shenzhen, China), allowing the establishment of all printing parameters (support type, layer exposure time, printing angle, and layer thickness). After saving the parameters, the samples were printed (Resilab Temp 3D Lot 1582) using the Anycubic printer at 120° relative to the horizontal plane, cleaned in isopropyl alcohol for 4 min in an ultrasonic bath, dried with absorbent paper, and treated in a post-curing chamber (Anycubic). Different post-curing times (non-additional post-curing, 16 min, 32 min, and 60 min) were applied for FT-IR analysis, Knoop microhardness, color and translucency analysis, flexural strength, and scanning electron microscopy (SEM). The remaining tests were conducted with a standardized post-curing time of 1 h.

### 2.2. FT-IR-Spectroscopy (Fourier-Transform-Infraroodspectroscopie)

The samples were printed in bar format with dimensions of 25 × 10 × 3 mm and cured for different times (no post-curing, 16 min, 32 min, and 60 min). Immediately after curing, the samples were cleaned with isopropyl alcohol to remove residual monomers and dried with absorbent paper. The polishing of the samples was standardized and carried out using rubber points with three different grits (coarse, medium, and fine) and a felt disc for final polishing using an electric motor. Immediately after polishing, the samples were sectioned. In this way, FT-IR readings were taken at different points on the samples (surface and center), in addition to readings of the liquid material for baseline analysis.

The spectrum was recorded in absorbance mode using a diamond crystal plate and obtained with a resolution of 4 cm^−1^ in the spectral range of 500–4000 cm^−1^. The experiment was conducted three times for each of the evaluated groups (Figure 1). In each spectrum, the absorption band heights of aliphatic and aromatic C=C bonds were measured at 1585 and 1785 cm^−1^, respectively. The degree of conversion (DC) was calculated using the formula described in the literature [19].

### 2.3. Scanning Electron Microscopy

The samples were cleaned in isopropyl alcohol in an ultrasonic bath and dried with absorbent paper, then packaged in gauze without contact with the operator’s hands to avoid contamination of the surface. After preparation, the samples were gold-sputtered. Four specimens were observed under the microscope (uncured, 16 min, 32 min, and 1 h of curing).

### 2.4. Three-Point Bending Flexural Strength

The samples (n = 9 for each group) were designed in Rhinoceros 6.0 and printed in the dimensions described in ISO 4049 [20] (25 × 10 × 3 mm) on the W3D printer from Wilcos (Wilcos do Brasil—Petrópolis, Brazil). They were positioned on a specific bending device (Figure 2) at a distance of 20 mm, and a load of 100 Kgf was applied until the specimen fractured. The values were obtained in N. The flexural strength (σ) was calculated in megapascals following Wendler et al. (2017) [17].

### 2.5. Knoop Microhardness

For microhardness analysis, a Knoop indenter was used in the microhardness tester (Shimadzu HMV-G21DT, Shimadzu, Kyoto, Japan). The specimens were 12 × 12 mm squares printed on the Anycubic printer following the standards described in second table in Section 3. In the test, a load of 300 g was applied for 15 s [17], and three indentations were made at three points on the surface. The test was repeated after the hydrothermal aging of samples carried out in a thermocycler (Biopdi Termocycle, Biopdi, São Carlos, Brazil), in which two thousand cycles were performed in baths of 30 s in water at 55 degrees and baths of 30 s in water at 5 degrees. The obtained values were evaluated by one-way ANOVA and Tukey 95%.

### 2.6. Color and Translucency Analysis

For color analysis, 3D files in the form of 12 × 12 cm disks were created in Rhinoceros software version 6.0. The file was exported in STL format, and in the printer slicing software, printing patterns were determined (described in second table in Section 3). In this way, 40 disks were printed. After printing, the samples were cleaned with isopropyl alcohol and cured in a photopolymerizer for different times (no cure [control], 16 min, 32 min, and 1 h). Twenty-four hours after curing, two readings were taken on each disk in the spectrophotometer (Easy shade, VITA Zahnfabrik, Bad Säckingen, Germany), against black, white, and gray backgrounds, to assess color and translucency based on the CIELab system. After the readings, the samples were individually packaged, labeled, and aged in a thermocycler (Biopdi Termocycle) for two thousand cycles in baths of 30 s in water at 55 degrees and baths of 30 s in water at 5 degrees. The values found before and after aging were treated using the formula below:$$\begin{array}{l} \Delta E^{\prime }=[{\left(\frac{\Delta L^{\prime }}{K_{L}S_{L}}\right)}^{2}+{\left(\frac{\Delta C^{\prime }}{K_{C}S_{C}}\right)}^{2}+{\left(\frac{\Delta H^{\prime }}{K_{H}S_{H}}\right)}^{2}\\ \;\;\;+R_{T}(\frac{\Delta C^{\prime }}{K_{C}S_{C}})(\frac{\Delta H^{\prime }}{K_{H}S_{H}}){]}^{1/2} \end{array}$$

### 2.7. Statistical Analysis

Statistical analysis was performed using the software Minitab version 16.1. The normality of the flexural strength data was assessed through the Shapiro–Wilk test, confirming a normal distribution of the strength test data (*p* > 0.05). Subsequently, one-way analysis of variance (ANOVA), followed by Tukey’s test (α = 0.05), were conducted to compare statistically significant differences among the groups for flexural strength. Similarly, the Shapiro-Wilk normality test indicated normal distribution for Knoop hardness and optical properties (*p* < 0.05). The values obtained for Knoop hardness and optical properties underwent statistical analysis using two-way ANOVA and Tukey’s 95%. Furthermore, a qualitative results presentation of the FT-IR and SEM results was performed.

## 3. Results

The data obtained from the FT-IR readings were tabulated and statistically analyzed using Origin software version 9.85. The chemical structure and corresponding FT-IR spectra of the individual monomers, with methacrylate peak assignments, are shown in Figure 3. Based on the spectrum of the band used, it was possible to characterize the chemical elements present in the material following Delgado and Young (2021) [21]. 

From the mean values, it was observed that the 60 min group showed a 46% degree of conversion on the surface, while the center of the 60 min samples had 37%. The 32 min groups exhibited 36% conversion on the surface and 29% in the center of the sample, the 16 min groups had 27% conversion on the surface and 19% in the center, and the no-cure group had 16% on the surface and 12% in the center (Figure 4).

The samples were analyzed using a scanning electron microscope to observe the surface characteristics. The images correspond to the surface of each specimen at different post-curing times (Figure 5). It is possible to observe that samples that were not post-cured, and those that were cured for 16 and 32 min, had surface pores (red arrows), while samples cured for 60 min had a more homogeneous surface. The yellow arrows indicate the silica particles present, measuring approximately 1 µm.

The obtained flexural strength values were subjected to statistical analysis using one-way ANOVA and Tukey’s 95% Test. The groups exhibited statistically significant differences (*p* = 0.001), as detailed in Table 1.

The Knoop hardness values obtained were subjected to statistical analysis using two-way ANOVA and Tukey’s 95%. The results are detailed in Table 2. While statistical differences were observed between polymerization times (*p* = 0.007) and aging (*p* = 0.001), it is important to note that pairwise comparisons reveal similarities between the groups before and after aging (Table 2).

In terms of optical properties, a two-way ANOVA and Tukey’s 95% test were conducted, indicating no statistically significant differences between the groups regarding variations in translucency (*p* = 0.373) and color change (*p* = 0.855) (Table 3).

## 4. Discussion

Additive manufacturing is gaining increasing prominence in dentistry, given significant advancements in printing technologies and materials. Therefore, understanding the properties of polymers used for prostheses and other structures is crucial to ensure effectiveness and quality. This study evaluated the mechanical and optical characteristics of a temporary printing polymer-based composite material.

Mechanical properties of printing resins have been relatively understudied, considering the vast array of commercially available materials, the number of technologies, and the recent implementation and clinical use of additive manufacturing. The technology used in this study was direct light processing (DLP), which is widely available commercially and clinically used. The chosen material was Resilab Temp (Wilcos do Brasil, Petrópolis, Brazil), a polymer-based composite material recommended for temporary prosthesis fabrication.

Several factors, such as printing parameters, layer thickness, slicing, printing orientation, material composition, and post-processing curing parameters, directly affect the final product [22]. Therefore, this study assessed the influence of three different post-curing times on mechanical analyses. In the study by Borella et al. [23], mechanical properties were analyzed based on two different printed layer thicknesses (50 and 100 mm) using four printing materials, including Resilab Temp. Drawing a comparation to the parameters evaluated in the present study, such as scanning electron microscopy, the authors note that when printed with a thickness of 100 mm, Resilab presents more surface defects compared to 50 mm, as used in the study. Another factor evaluated was Vickers microhardness, which showed the lowest index among the groups for Resilab Temp. This was attributed to the fact that it was a resin indicated for temporary use, whereas the others used were resins filled with inorganic particles, resulting in better mechanical performance compared to Resilab Temp.

Based on the results, the post-curing times with the best performance for flexural strength were 16 min and 60 min, with strengths exceeding 80 MPa. Similar values were found in other studies comparing conventional, milled, and printed polymers, which conducted flexural strength and surface hardness tests.

The flexural strengths of the evaluated resins were as follows: Nextdent Base with 84.5 MPa, NextDent’s Ortho Rigid with 75 MPa, and Bego’s Varseo Wax with a superior value of 117.2 MPa. These results are related to resin compositions. Varseo Wax from Bego, for example, has a higher resistance value due to the addition of ceramic nanoparticles. Although the other brands have similar values, their clinical indicators differ from the resin used in this study, and the study did not provide information on printing patterns and printer types, preventing direct comparisons [24,25]. Additionally, the comparisons that can be made between different studies are limited since different geometries have been used during specimen manufacturing. Only by using the Weibull statistic can an equivalent be calculated, considering Weibull modulus and the reliability of the data distribution. In the present study, 32 min post-curing showed an intermediate influence on the material’s strength. One possible explanation is that even though the resin was cured for 32 min, it is possible that it did not fully cure during that time. Factors such as insufficient curing light intensity or improper curing conditions could result in incomplete curing and weaker final properties. This theory is based on the fact that the resin cured for 60 min exhibited greater strength than the one cured for 32 min. Finally, the group cured for 16 min is expected to be less cured and therefore softer than those cured for 32 min and 60 min. The incomplete cure was confirmed by FT-IR results. With that in mind, it is known that soft materials often possess higher ductility, meaning they can deform significantly before failure. This deformation absorbs energy and can increase the material’s ability to withstand loads without breaking. This could suggest why materials that underwent 16 min of curing time showed higher flexural strength than those that underwent 32 min of curing.

Regarding surface hardness, the mentioned resins had different values, ranging from 15.5 MPa for NextDent Base (15 min of post-curing time) to 28.5 MPa for Varseo Wax [26,27]. This value is similar to the Knoop microhardness result for the polymer-based composite material used in this study, with 28.4 MPa. This hardness value decreased after sample aging, consistent with other studies. Soto-Montero et al. [24] compared four types of printing resins (Resilab Temp (Wilcos do Brasil, Petrópolis, Brazil), Cosmos (Yller, Pelotas, Brazil), Prizma (PriZma 3D Bio, MarketechLabs, São Paulo, Brazil), and Smart Print (SMA Tech, Tabocas do Brejo, Brazil)*,* all of which had been indicated as suitable for temporary restorations. The study achieved better results for Resilab Temp and Prizma with 15 min of curing.

Factors such as construction orientation and printing angle affect material properties, product accuracy, and even biocompatibility. The printing angle used for sample manufacturing in this study was 120 degrees, following Park et al. [2], who showed that angles of 135 degrees and 120 degrees reduce internal gaps, increasing adaptation and stability. Osman et al. (2017) stated that the 135-degree angle is best for DLP printers, enhancing accuracy [28]. Meanwhile, Alharbi et al. (2016), using different angles in molar crowns, claimed that the 120-degree angle provides greater dimensional precision and requires less support surface in crowns [29].

The degree of conversion is crucial for investigating resin mechanical performance and biocompatibility since it is directly related to material fracture resistance, hardness, and solubility. A low resin conversion can result in free, not-reacted monomers that may dissolve in humid environments, leading to material degradation and compromising the restoration’s longevity. The degree of conversion (DC) obtained from FT-IR analysis in the best-performing group was 46% on the surface and 37% at the center of samples with 1 h of post-curing. In contrast, Bergamo et al. [30] found a conversion degree of 66.5% in a printing resin (Cosmos, Yller) with the same indication as the resin used in this study, with the conversion degree increasing by 5% after aging. However, it is essential to recognize that comparing the DC of materials with different compositions requires careful consideration of various factors, such as the type and concentration of monomers, the presence of additives, and the curing conditions. All of these factors can influence the polymerization kinetics and final DC.

The optical properties of the temporary material must be satisfactory, especially for restorations used in the aesthetic sector, requiring stability and resistance to possible color changes when exposed to the oral environment, which is constantly in contact with different substances and temperatures. This study conducted color and translucency analyses of Resilab Temp both immediately and following post-processing curing for different lengths of time (60 min, 32 min, 16 min, and no post-curing), and after thermal aging.

This study revealed no difference between post-curing times regarding color change and translucency. A previous study [24] evaluated four different types of printing resin (Cosmos (Yller), Prizma (3D Bio Prov), Smart Print (SMA Tech), and Resilab Temp (Wilcos do Brasil)) using a methodology similar to ours, comparing curing times (0, 5, 15, and 20 min) and obtained different results regarding color change, comparing time and resins. Analyzing isolated curing times and the resin used in our study (Resilab temp), we noted similar results, showing stability in this material. Regarding translucency, the authors mentioned differences between resins, but did not indicate differences between post-curing times for each resin.

There is a reference value in the literature regarding acceptability and perceptibility levels for color changes. Paravina et al. [31] define that ΔE00 values of 0.80 refer to a 50:50% perceptibility level, and ΔE00 of 1.80 refers to a 50:50% acceptability level. Following these reference values, all groups are within the acceptability level, except the aged 32 min group. However, only the 1 h group is within the perceptibility level. It is essential to reiterate that this study used CIEDE2000, which corrects CIElab, improving acceptability and perceptibility determination. In addition, it would be pertinent to consider the influence of association printing parameters, such as layer thickness and curing time, on the mechanical properties of the 3D-printed resin, as well as the possible impact of alternative post-curing methods [32].

In contrast to a prior investigation that examined the influence of nanoparticle addition and post-curing time (PCT) on the flexural properties of 3D-printed denture base resins, our study exclusively concentrated on evaluating the effect of post-curing time on the microstructure, optical properties, and mechanical behavior of a temporary polymer-based composite material [26]. However, the present study corroborates with the aforementioned authors, indicating that the flexural strength and hardness of a material can be optimized with longer post-curing times. Another difference was that all the specimens from the study from AlGhamdi et al. [26] were subjected to thermal stress before testing to simulate the effects of aging, while herein half of them were immediately tested and the other half were tested after thermal aging. With this approach, it was possible to observe that simulated aging dampened the measured properties. The aging effect of resinous materials can result from factors such as polymer chain scission, filler-matrix debonding, and water sorption, leading to plasticization or hydrolysis of the polymer matrix.

Studies into properties such as tensile strength, impact resistance, fracture toughness, wear resistance, and fatigue testing are important for mechanical characterization and also for evaluating biomechanical behavior. These are in addition to studies involving water absorption, solvent resistance, temperature of post-curing processing, and biocompatibility, which are also important for the structural characterization of materials and long-term behavior in oral environments.

## 5. Conclusions

Based on the obtained results, it is concluded that additional curing time significantly influences the mechanical properties of the temporary polymer-based composite material examined in this study. Thermal aging induces alterations in the resin properties, causing a decrease in hardness with thermocycling, while maintaining color stability. Flexural strength is directly influenced by curing time, with the highest value observed after 1 h of curing. The Resilab Temp printing resin (Wilcos) is deemed satisfactory for temporary prosthetic restorations. 

## Figures and Tables

**Figure 1 materials-17-01496-f001:**
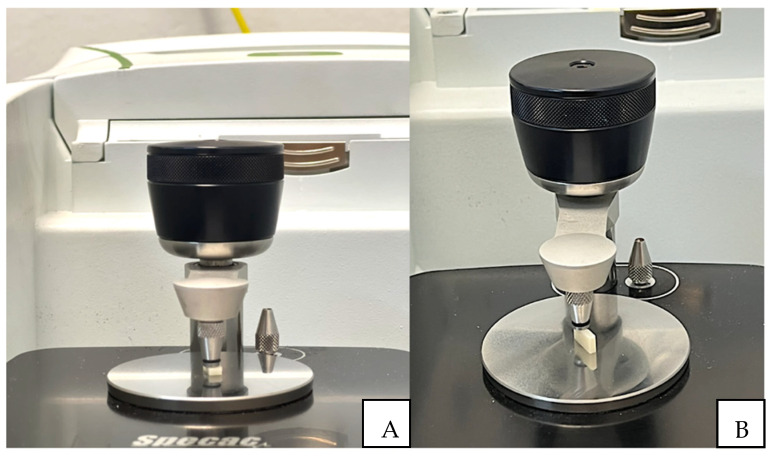
(**A**) Specimen positioned on the FT-IR for surface reading; (**B**) Specimen positioned on the FT-IR for reading of the central area.

**Figure 2 materials-17-01496-f002:**
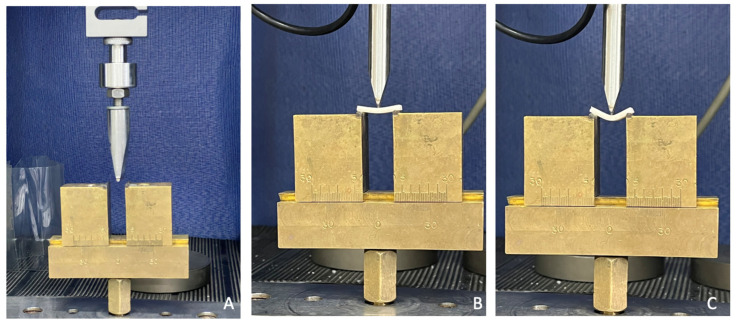
A universal testing machine configured for the flexural strength test. (**A**) Specimen positioned for the start of the bending test; (**B**,**C**) Images showing the deformation of the specimen before fracture.

**Figure 3 materials-17-01496-f003:**
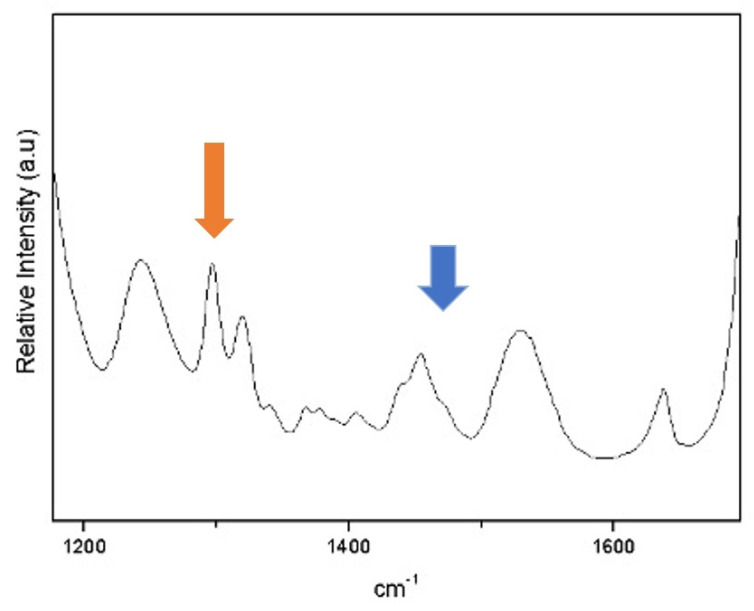
The peaks represented by the orange arrow correspond to the band with the presence of TEGDMA, and the peak with the blue arrow represents the UDMA band, according to Delgado and Young (2021) [21].

**Figure 4 materials-17-01496-f004:**
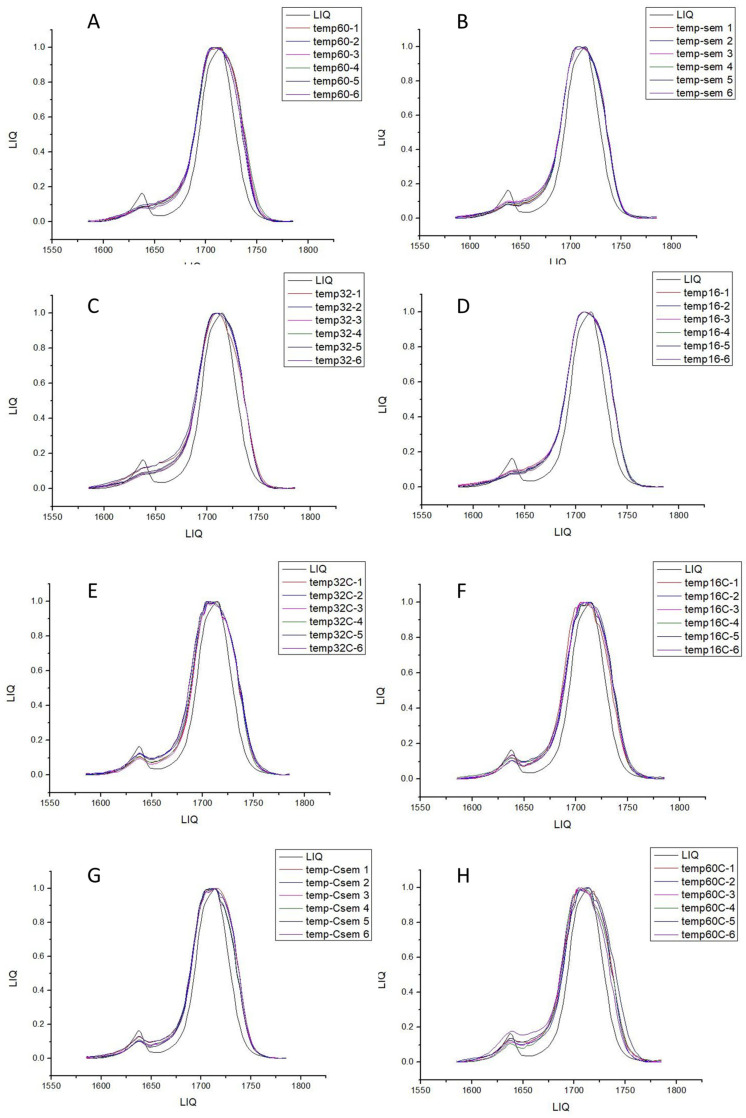
Representative graph of the peak between the liquid and uncured sample readings from the center and surface of the specimen, followed by figures (**A**–**H**), with peak representative graphs for 16 min, 32 min surface, 60 min surface, 16 min center, 32 min center, and 60 min center, respectively.

**Figure 5 materials-17-01496-f005:**
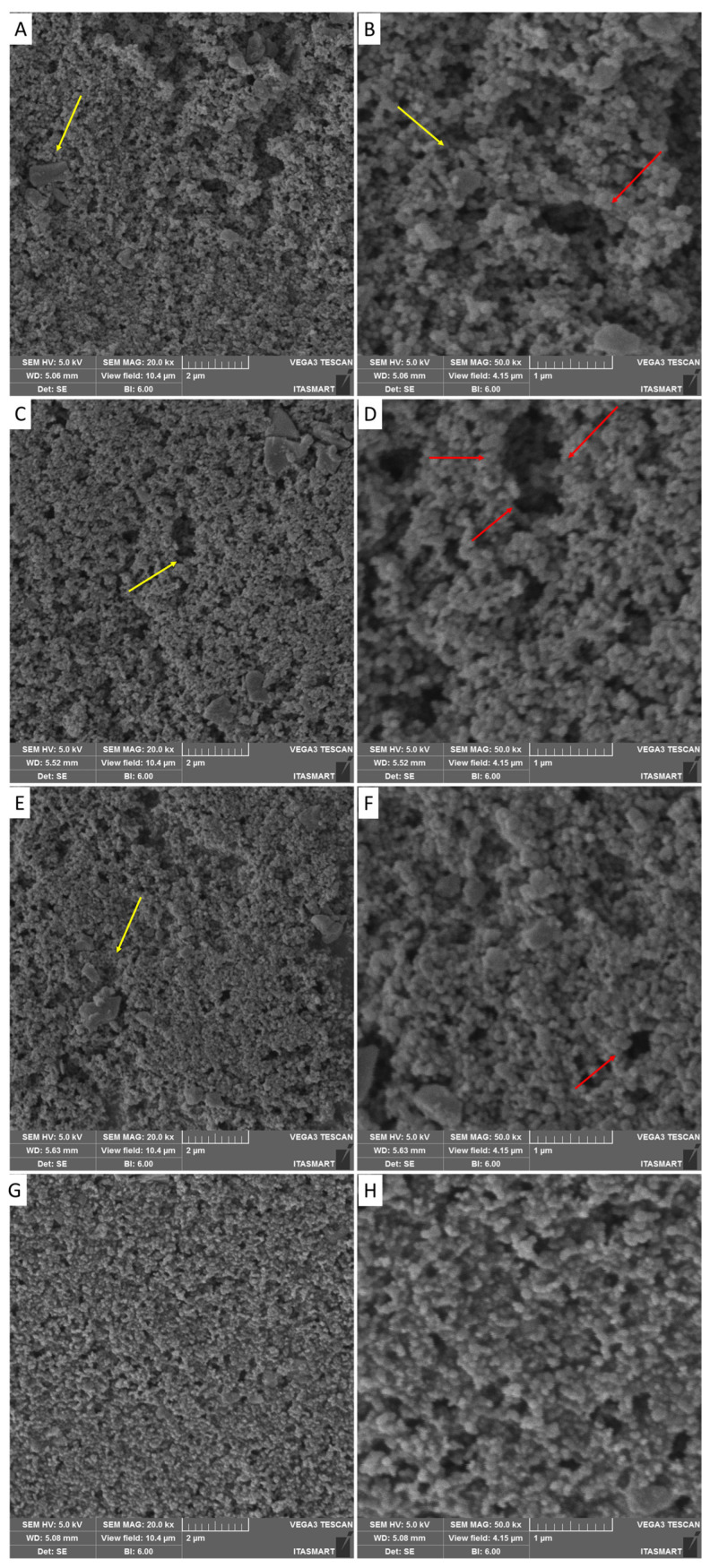
(**A**,**B**) Scanning electron microscopy of the surface of the non-post cure temporary resin, magnification of 20K× and 50K× (**C**,**D**) scanning electron microscopy of the surface of the temporary resin with 16 min of curing, magnification of 20K× and 50K× respective; (**E**,**F**) scanning electron microscopy of the surface of the temporary resin with 32 min of curing magnification of 20K× and 50K×; (**G**,**H**) scanning electron microscopy of the surface of the temporary resin with 60min of curing, magnification of 20K× and 50K×.

**Table 1 materials-17-01496-t001:** Flexural strength according to different post-curing times (average values and standard deviation).

Group	Average ± SD (MPa)	Tukey
Control	15.9 ± 3.8	C
16 min	80.5 ± 3.2	A
32 min	76.5 ± 1.6	B
60 min	83.2 ± 2.2	A

**Table 2 materials-17-01496-t002:** Knoop microhardness average values, according to aging and curing time.

Aging	Curing Time	Average (HV)	Tukey
No	Control (non post-cured)	24.7 ± 5.2	A B C
	16 min	23.6 ± 3.2	A B C
	32 min	26.6 ± 6.1	A B
	60 min	28.4 ± 6.2	A
Yes	Control (non post-cured)	20.1 ± 1.0	C
	16 min	22.0 ± 1.7	B C
	32 min	22.5 ± 1.8	A B C
	60 min	25.3 ± 3.5	A B C

**Table 3 materials-17-01496-t003:** Optical properties analysis: translucency variation and color change among groups.

Aging	Curing Time	Translucency	Color (ΔE_00_)
No	Control (non post-cured)	8.23 ± 1.24	1.05 ± 0.60
	16 min	7.98 ± 1.61	1.21 ± 0.55
	32 min	7.01 ± 2.08	1.02 ± 0.33
	60 min	7.98 ± 1.61	0.42 ± 0.26
Yes	Control (non post-cured)	6.84 ± 2.24	1.67 ± 0.89
	16 min	8.31 ± 1.68	1.09 ± 0.42
	32 min	7.49 ± 1.20	2.10 ± 1.49
	60 min	6.81 ± 1.38	0.92 ± 1.06

## Data Availability

Data is available upon reasonable request from the first author.

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
