# Peer review of "Characterization of Microstructure, Optical Properties, and Mechanical Behavior of a Temporary 3D Printing Resin: Impact of Post-Curing Time"

_materials, 2024, doi:10.3390/ma17071496_

Round 1

Reviewer 1 Report

Comments and Suggestions for Authors

The manuscript with the title "Characterization of Microstructure, Optical Properties, and Mechanical Behavior of a Temporary 3D Printing Resin: Impact of 3 Post-Processing Curing Time"

Manuscript ID: materials-2866383

Dear Authors,

In the beginning, I would like to express words of my appreciation for the idea and effort put into conducting research and writing the manuscript recommended to me for review.

General Comments:

The authors characterized the microstructure of a temporary 3D printing resin, evaluated the mechanical and optical properties as a function of post-processing curing time. Characterization of the prints was carried out using Fourier Transform Infrared Spectroscopy, Scanning Electron Microscopy, flexural strength tests and Knoop microhardness. Color and transparency measurements were also included. The results obtained were subjected to statistical analysis

Specific Comments:

The work is appropriate for the journal but there are some observations that must be addressed before the manuscript can be accepted.

1)    The term in line 122 "polymerized in a photopolymerizer (Anycubic)" is slightly confusing. Why polymerize a polymerized print? I would suggest using the phrase, for example, "treated in post-curing chamber" or similar

2)    Figure 3 is unreadable. I would suggest changing the way the results are presented

3)    Letter designations in Figure 4 showing SEM images include highlights from the word processor used by the authors. It would be good to remove them

4)    There is no description of the differences seen in the SEM images and the possible reason for their occurrence

5)    The explanation of lower flexural strength for 32 min post-curing is missing. This relationship deviates strongly from the expected result. Similarly for the relationship in Table 2 for unaged samples

6)    Some of the tables contain centered text and some do not. I would suggest introducing consistent editing of tables

7)    Typo in tables  "1 hora"

8)     Conversions should be shown how they were calculated by the authors. In FTIR spectroscopy, we are not talking about peaks but the corresponding bands. How the baseline was calculated, whether the band area or intensity was taken into account.

Final State of the review:

The work presented for review shows the exciting work of authors that was put into this manuscript. The article, as research work, meets the criteria set for authors to be published in materials.

Therefore, I ask the Editor to recommend this work for publication after MINOR REVISION.

Comments on the Quality of English Language

The manuscript with the title "Characterization of Microstructure, Optical Properties, and Mechanical Behavior of a Temporary 3D Printing Resin: Impact of 3 Post-Processing Curing Time"

Manuscript ID: materials-2866383

Dear Authors,

In the beginning, I would like to express words of my appreciation for the idea and effort put into conducting research and writing the manuscript recommended to me for review.

General Comments:

The authors characterized the microstructure of a temporary 3D printing resin, evaluated the mechanical and optical properties as a function of post-processing curing time. Characterization of the prints was carried out using Fourier Transform Infrared Spectroscopy, Scanning Electron Microscopy, flexural strength tests and Knoop microhardness. Color and transparency measurements were also included. The results obtained were subjected to statistical analysis

Specific Comments:

The work is appropriate for the journal but there are some observations that must be addressed before the manuscript can be accepted.

1)    The term in line 122 "polymerized in a photopolymerizer (Anycubic)" is slightly confusing. Why polymerize a polymerized print? I would suggest using the phrase, for example, "treated in post-curing chamber" or similar

2)    Figure 3 is unreadable. I would suggest changing the way the results are presented

3)    Letter designations in Figure 4 showing SEM images include highlights from the word processor used by the authors. It would be good to remove them

4)    There is no description of the differences seen in the SEM images and the possible reason for their occurrence

5)    The explanation of lower flexural strength for 32 min post-curing is missing. This relationship deviates strongly from the expected result. Similarly for the relationship in Table 2 for unaged samples

6)    Some of the tables contain centered text and some do not. I would suggest introducing consistent editing of tables

7)    Typo in tables  "1 hora"

8)     Conversions should be shown how they were calculated by the authors. In FTIR spectroscopy, we are not talking about peaks but the corresponding bands. How the baseline was calculated, whether the band area or intensity was taken into account.

Final State of the review:

The work presented for review shows the exciting work of authors that was put into this manuscript. The article, as research work, meets the criteria set for authors to be published in materials.

Therefore, I ask the Editor to recommend this work for publication after MINOR REVISION.

Author Response

Dear Authors,

In the beginning, I would like to express words of my appreciation for the idea and effort put into conducting research and writing the manuscript recommended to me for review.

General Comments:

The authors characterized the microstructure of a temporary 3D printing resin, evaluated the mechanical and optical properties as a function of post-processing curing time. Characterization of the prints was carried out using Fourier Transform Infrared Spectroscopy, Scanning Electron Microscopy, flexural strength tests and Knoop microhardness. Color and transparency measurements were also included. The results obtained were subjected to statistical analysis

Specific Comments:

The work is appropriate for the journal but there are some observations that must be addressed before the manuscript can be accepted.

  • The term in line 122 "polymerized in a photopolymerizer (Anycubic)" is slightly confusing. Why polymerize a polymerized print? I would suggest using the phrase, for example, "treated in post-curing chamber" or similar

R: We accepted your suggestion and the phrase was updated.

  • Figure 3 is unreadable. I would suggest changing the way the results are presented

R:  The images were changed and the explanation was made in the legend.

  • Letter designations in Figure 4 showing SEM images include highlights from the word processor used by the authors. It would be good to remove them

R: Figure 4 was updated using arrows to point to the details, and the letters were also modified.

  • There is no description of the differences seen in the SEM images and the possible reason for their occurrence

R: It was added a simple explanation regarding the SEM in the results section, and a better explanation on the discussion section.

  • The explanation of lower flexural strength for 32 min post-curing is missing. This relationship deviates strongly from the expected result. Similarly for the relationship in Table 2 for unaged samples

R: One possible explanation is that even though the resin was cured for 32 minutes, it's possible that it didn't fully cure during that time. Factors such as insufficient curing light intensity or improper curing conditions could result in incomplete curing and weaker final properties. This theory is based on the fact that the resin cured for 60 minutes exhibited greater strength than the one cured for 32 minutes. Finally, the group cured for 16 minutes is expected to be less cured and therefore softer than those cured for 32 minutes and 60 minutes. The incomplete cure was confirmed by FTIR results. With that in mind, it is known that soft materials often possess higher ductility, meaning they can deform significantly before failure. This deformation absorbs energy and can increase the material's ability to withstand loads without breaking. This would maybe suggest why 16 minutes of curing time showed higher flexural strength than 32 minutes.

  • Some of the tables contain centered text and some do not. I would suggest introducing consistent editing of tables

R: The tables were edited.

  • Typo in tables  "1 hora"

R: This information was corrected.

  • Conversions should be shown how they were calculated by the authors. In FTIR spectroscopy, we are not talking about peaks but the corresponding bands. How the baseline was calculated, whether the band area or intensity was taken into account.

R: A new figure (new 3) and an explanation were added regarding the FT-IR.

Reviewer 2 Report

Comments and Suggestions for Authors

The study by Joyce Roma Correia dos Santos Siqueira et al discusses the effect of light post curing on a resin for UV 3D printing. The study misses some key information and needs to be revised. 

For example, the paper discusses a resin called "resilab temp (wilcos do brasil)". We can only assume that it is the product found on the following link: https://wilcos.com.br/catalogo/detalhes/563 . The authors do not provide any information regarding the material, and the site also does not provide any information nor any MSDS or TDS files to try and extract some info from. Is it an aliphatic acrylate, an urethane acrylate, a bisphenol-A type acrylate, methacrylate, epoxy chemical? How can the authors reach any valuable conclusions without knowing? And also how can they compare this resin with the previous studies they mention in the literature (citations 20,21)? Finally, this material is the main material of the study, and it is mentioned "in detail" for the first time in page 9!

Regarding the curing of the material, the authors in many occasions in the text refer to the printed samples as uncured. This term is very confusing for the reader, as printed samples are UV cured. Otherwise they would be still in resin form and not in free-standing bars. The authors should rephrase it as non post-cured or a similar term of their choice.  

On the matter of the printed samples, the preparation for the FTIR measurements is not clear. For the "center" measurement, do they measure a cross-section of the material? in the pictures provided it seems that they simply measure at different point of the surface of the printed bar. And if that is the case, how can the authors explain that on a DLP printer that prints in a layer by layer fashion there are differences in the conversion on the same layer? There is no logical explanation why there is a difference on the C=C conversion, unless the resin/formulation is not homogenous, which we dont know since the authors do not provide any useful information on the resin...

In the Tables provided there are on many occasions terms that are not in English (hora, sem cura etc). Please write the whole manuscript in English. 

From the mechanical properties of the materials, we can see that the post curing time of 16 mins provides the same properties with the 1 hour post curing (flexural strength 80 MPa at 16 min, 83 MPa at 1 hour/ microhardness 24 HV to 28 HV). What is the reason behind this extensive post curing times? It seems that it would be more useful to study post curing times shorter than 16 mins, to determine the minimum time needed for the maximum improvement on the mechanical properties. 

The authors should include the fact that the samples printed at 120 degrees in the experimental part. 

Regarding the conversion from the FTIR, the authors claim that they used the same protocol as in reference 18. There the reference signal is an aromatic signal at 1600 cm-1. In this study there is no such signal. Which signal did the authors use? If they used the possibly ester signal at 1730 cm-1 intergration with origin software would be rather difficult, as the C=C signal appears as a shoulder of the main signal at 1730 cm-1. If indeed the resin is an acrylate of some sort, you should better try the signal of the C=C around 800 cm-1. Probably the results will be much different

Comments on the Quality of English Language

Minor editing of English language required

Author Response

The study by Joyce Roma Correia dos Santos Siqueira et al discusses the effect of light post curing on a resin for UV 3D printing. The study misses some key information and needs to be revised. 

For example, the paper discusses a resin called "resilab temp (wilcos do brasil)". We can only assume that it is the product found on the following link: https://wilcos.com.br/catalogo/detalhes/563 . The authors do not provide any information regarding the material, and the site also does not provide any information nor any MSDS or TDS files to try and extract some info from. Is it an aliphatic acrylate, an urethane acrylate, a bisphenol-A type acrylate, methacrylate, epoxy chemical? How can the authors reach any valuable conclusions without knowing? And also how can they compare this resin with the previous studies they mention in the literature (citations 20,21)? Finally, this material is the main material of the study, and it is mentioned "in detail" for the first time in page 9!

R: Thank you for your revision. The resin used in this study was Resilab Temp by Wilcos, however, as the reviewer said, the company does not provide information about the material. However, a previous characterization of this resin was carried out and added to the figure 3

Regarding the curing of the material, the authors in many occasions in the text refer to the printed samples as uncured. This term is very confusing for the reader, as printed samples are UV cured. Otherwise they would be still in resin form and not in free-standing bars. The authors should rephrase it as non post-cured or a similar term of their choice.  

R: We agreed with the reviewer, and included in the text expression as: no additional post-curing, non post-cured, etc., were added.

On the matter of the printed samples, the preparation for the FTIR measurements is not clear. For the "center" measurement, do they measure a cross-section of the material? in the pictures provided it seems that they simply measure at different point of the surface of the printed bar. And if that is the case, how can the authors explain that on a DLP printer that prints in a layer by layer fashion there are differences in the conversion on the same layer? There is no logical explanation why there is a difference on the C=C conversion, unless the resin/formulation is not homogenous, which we dont know since the authors do not provide any useful information on the resin...

R: The samples were cut after post-curing and measurements were taken at the surface and in the center (Images have been added for better visualization). Although the printer prints in layers, additional polymerization is carried out, which can improve the degree of surface conversion and may not have the same effect on the center of the material.

In the Tables provided there are on many occasions terms that are not in English (hora, sem cura etc). Please write the whole manuscript in English. 

R: Ok, it was corrected.

From the mechanical properties of the materials, we can see that the post curing time of 16 mins provides the same properties with the 1 hour post curing (flexural strength 80 MPa at 16 min, 83 MPa at 1 hour/ microhardness 24 HV to 28 HV). What is the reason behind this extensive post curing times? It seems that it would be more useful to study post curing times shorter than 16 mins, to determine the minimum time needed for the maximum improvement on the mechanical properties. 

R: As stated in the discussion of this work, the study by Soto-Montero et al. (2022) showed that times shorter than 15 min are insufficient.

The authors should include the fact that the samples printed at 120 degrees in the experimental part. 

R: It was updated.

Regarding the conversion from the FTIR, the authors claim that they used the same protocol as in reference 18. There the reference signal is an aromatic signal at 1600 cm-1. In this study there is no such signal. Which signal did the authors use? If they used the possibly ester signal at 1730 cm-1 intergration with origin software would be rather difficult, as the C=C signal appears as a shoulder of the main signal at 1730 cm-1. If indeed the resin is an acrylate of some sort, you should better try the signal of the C=C around 800 cm-1. Probably the results will be much different

R: In the -800 region several band overlaps are characteristic of carbon-oxygen bonds and charges such as silica itself present in the material, these band overlaps do not allow the observation of any -800 band in isolation, hence the use of carbonyl band from 1600 to 1700 as reference.

Reviewer 3 Report

Comments and Suggestions for Authors

The paper examines the impact of post-processing curing time on the mechanical, microstructural, and optical properties of a temporary 3D printing resin named Resilab Temp.

The paper is original in its main focus and about the different processes affecting the material's performance, particularly for dental applications.

The study is well-structured, employing a range of analytical techniques such as FTIR spectroscopy, scanning electron microscopy, and mechanical testing to assess the resin's properties. The findings suggest that a post-polymerization time of 1 hour can improve mechanical performance, which is promising for the development and optimization of 3D-printed dental devices. The experimental section is valuable, a strong starting point in the positive evaluation of a paper, in my opinion.

A more comprehensive understanding of the results and a closer look at the unique benefits and constraints of Resilab Temp could be obtained by comparing its performance with that of other temporary printing resins under equivalent testing circumstances. To improve the paper significantly, I would say, the comparative discussion with other papers in the field needs to be expanded to cover the impact of printing parameters on mechanical properties to the effects of composition, fabrication method, and aging on mechanical properties. The improvements of the Resilab Temp introduced in this paper may be better understood by extending the connections with research literature in particular the reference 24.

I am aware that to provide a more thorough understanding of the mechanical properties of the resin, further mechanical testing beyond flexural strength and Knoop microhardness, such as tensile strength, impact resistance, and fatigue testing, will be required for future studies. To determine whether the resin is suitable for long-term clinical use, more research on durability (water absorption, solvent resistance), wear resistance, and biocompatibility testing may be helpful.

One further suggestion: please standardize the terminology. Post-processing time should be used consistently throughout the document to ensure clarity and prevent confusion with other processes like thermal aging or post-curing, unless they are specifically mentioned as distinct steps within the post-processing phase. Alternatively, if definitions require differentiated nuances, these should be explained.

The work is the result of a valuable research that follows a clear methodology and results presentation. I suggest its publication in Materials - surely a pertinent magazine - after extended comparisons as suggested above.

Comments on the Quality of English Language

I only asked to look at the consistency of terminology

Author Response

The paper examines the impact of post-processing curing time on the mechanical, microstructural, and optical properties of a temporary 3D printing resin named Resilab Temp.

The paper is original in its main focus and about the different processes affecting the material's performance, particularly for dental applications.

The study is well-structured, employing a range of analytical techniques such as FTIR spectroscopy, scanning electron microscopy, and mechanical testing to assess the resin's properties. The findings suggest that a post-polymerization time of 1 hour can improve mechanical performance, which is promising for the development and optimization of 3D-printed dental devices. The experimental section is valuable, a strong starting point in the positive evaluation of a paper, in my opinion.

R: Thank you for your revision.

A more comprehensive understanding of the results and a closer look at the unique benefits and constraints of Resilab Temp could be obtained by comparing its performance with that of other temporary printing resins under equivalent testing circumstances. To improve the paper significantly, I would say, the comparative discussion with other papers in the field needs to be expanded to cover the impact of printing parameters on mechanical properties to the effects of composition, fabrication method, and aging on mechanical properties. The improvements of the Resilab Temp introduced in this paper may be better understood by extending the connections with research literature in particular the reference 24.

R: Another study was included to reinterview the influence of parameters on the properties of the material, however, the literature is still scarce regarding the material mentioned in the present study.

I am aware that to provide a more thorough understanding of the mechanical properties of the resin, further mechanical testing beyond flexural strength and Knoop microhardness, such as tensile strength, impact resistance, and fatigue testing, will be required for future studies. To determine whether the resin is suitable for long-term clinical use, more research on durability (water absorption, solvent resistance), wear resistance, and biocompatibility testing may be helpful.

R: Thank you for your suggestion. We added this information on the discussion section.

One further suggestion: please standardize the terminology. Post-processing time should be used consistently throughout the document to ensure clarity and prevent confusion with other processes like thermal aging or post-curing, unless they are specifically mentioned as distinct steps within the post-processing phase. Alternatively, if definitions require differentiated nuances, these should be explained.

R: We have standardized the term non-post-curing.

The work is the result of a valuable research that follows a clear methodology and results presentation. I suggest its publication in Materials - surely a pertinent magazine - after extended comparisons as suggested above.

 R: Thank you so much.

Round 2

Reviewer 2 Report

Comments and Suggestions for Authors

The paper should be accepted after the changes made

Author Response

Thank you for your time and consideration for this manuscript.